# Delphinidin-3-rutinoside from Blackcurrant Berries (*Ribes nigrum*): In Vitro Antiproliferative Activity and Interactions with Other Phenolic Compounds

**DOI:** 10.3390/molecules28031286

**Published:** 2023-01-28

**Authors:** Bojana Miladinovic, Miguel Ângelo Faria, Mafalda Ribeiro, Maria Madalena Costa Sobral, Isabel M. P. L. V. O. Ferreira

**Affiliations:** 1Department of Pharmacy, Faculty of Medicine, University of Niš, Dr Zoran Djindjic blvd. 81, 18000 Niš, Serbia; 2LAQV/REQUIMTE, Departamento de Ciências Químicas, Laboratório de Bromatologia e Hidrologia, Faculdade de Farmácia—Universidade do Porto, 4050-313 Porto, Portugal

**Keywords:** antagonism, combination index, phenolic compounds, smart combinations, synergism

## Abstract

Blackcurrant berries (*Rigrum* L.) are of great interest for food scientists/technologists as a source of delphinidin-3-rutinoside (D3R). This is an uncommon phenolic compound in diets that unveils potent antiproliferative activity besides its colour. Other phenolic compounds, such as chlorogenic acid (CA) and epicatechin (EC), also known by their antiproliferative effects, are abundant in foods and beverages. To design smart food/supplements combinations containing blackcurrant and improved anticancer properties at the gastrointestinal level, there is the need for more data concerning the combined effects of those molecules. In this work, synergistic, additive, or antagonistic effects against gastric and intestinal cancers of D3R, CA, and EC were assessed in vitro. The antiproliferative activity of D3R, CA, and EC, alone and in binary combinations (D3R+CA, D3R+EC, and CA+EC) on NCI-N87 (gastric) and Caco-2 (intestinal) cells, was assessed following the Chou-Talalay theorem at equipotent contributions (i.e., (IC_50_)_1_/(IC_50_)_2_). D3R presented the strongest antiproliferative activity of the single molecules tested, with IC_50_ values of 24.9 µM and 102.5 µM on NCI-N87 and Caco-2 cells, respectively. The combinations D3R+CA and CA+EC were synergic against NCI-N87 until IC_50_ and IC_75_, respectively, while D3R+EC shifted from slight antagonism to synergism at higher doses. On Caco-2 cells, antagonism at low doses and synergism at high doses was observed. Therefore, the synergisms observed on the gastric cancer model at low doses occurred on the colon model only at high doses. Data herein described is vital to the targeted smart design of foods and supplements, as it is foreseen that the same combination of phenolic compounds causes different interactions/effects depending on the dose and gastrointestinal compartment.

## 1. Introduction

Gastric and colorectal cancers remain among the most common and deadly cancers worldwide [1]. Besides genetics and autoimmune (gastritis and/or ulcerative colitis)—or in the case of gastric cancer, bacterial (*Helicobacter pylori*)—causes, diet plays a relevant role in the development or prevention of those cancers [2,3,4]. Dietary polyphenols can affect and modulate multiple biochemical processes and pathways involved in carcinogenesis [5]. They can act as biological response modifiers supporting immune system function as well as protecting living cells against damage from free radicals [6]. Among the most abundant polyphenols in the human diet is epicatechin (EC), a flavanol commonly present in berries, apple, cocoa, and green tea [7], as well as chlorogenic acid (CA), a hydroxycinnamic acid naturally present in coffee beans, spices, fruits, including berries, vegetables, olive oil, and wine [8].

Delphinidin-3-rutinoside (D3R) is a rare polyphenol in the human diet that presents potent antiproliferative activity besides its intense black–purple colour [9,10,11,12,13,14]. Blackcurrant berries (*Ribes nigrum* L.) are a major source of D3R. This berry is trendy and widely accepted by consumers with a huge potential in terms of anticancer properties [10], being of great interest for food technologists (researchers and companies) to obtain smart combination of polyphenol rich foods that gather increased bioactive effects [14,15,16,17].

The described molecules (D3R, EC, and CA) act as cancer preventive agents, throughout several biochemical processes, which have been studied using both in vitro and in vivo models [10]. However, the vast majority of them were performed using individual compounds or complex extracts, without comprehensive exploitation of the combined effects/interactions occurring amongst molecules. Nonetheless, interactions between ingested phenolic compound mixtures occur at target organs and their effects can synergize with each other to elicit a higher antiproliferative impact (synergism) or be lower (antagonism) than the sum of individual compounds. Polyphenols interactions should be deeply studied as mixtures, since some combinations revealed antagonistic effects concerning their in vitro antioxidant response (e.g., 2,2-diphenyl-1-picrylhydrazyl (DPPH), free-radical scavenging ability, and ferric-reducing antioxidant power (FRAP) assays) [18]. This demonstrates that combining individually promising compounds may not necessarily confer an improved effect, i.e., synergistic. Moreover, the doses at which these compounds are present in diet may also determine the interaction (synergism/antagonism). This is a relevant matter in properly designing functional/bioactive foods as their joined contents can improve or worsen their bioactivity [19].

The aim of this study is to provide knowledge of the combined antiproliferative effects covering a high range of doses of D3R, and other polyphenols, such as EC and CA, against two widely used and well-characterized gastrointestinal cancers cell models representing two important compartments of the gastrointestinal system: the stomach—NCI-N87 gastric cells [20], and the intestine—Caco-2 intestinal epithelium [21]. The use of well-characterized in vitro cell culture systems offers an important strategy for studying cell responses to dietary compounds, leading to a better understanding of the in vivo response to diet and its role in cancer prevention.

## 2. Results

### 2.1. Antiproliferative Activity of Individual Compounds

The polyphenols D3R, EC, and CA exhibited different dose-effect responses on NCI-N87 and Caco-2 cells after individual exposure (Figure 1a). Table 1 exhibits the parameters concerning curve fit and shape, as well as inhibition concentration values (IC_25_, IC_50_, and IC_75_) concerning NCI-N87 and Caco-2 cells. To obtain IC_50_ values for all compounds, as required by the Chou-Talalay median-effect equation method carried out at an equipotency ratio, and since the dose-effect response differed between cell lines and within polyphenols, the curves were built using distinct ranges of concentrations (see Section 4.4.).

D3R was the compound with the strongest antiproliferative effect on both cell lines exhibiting the lowest IC_50_ values of 24.9 µM and 102.5 µM on NCI-N87 and Caco-2 cells, respectively. Based on the individual IC_50_ values, the phenolic compounds were ranked according to their antiproliferative effects as D3R (IC_50_ = 24.9 µM) > EC (IC_50_ = 126.1 µM) > CA (IC_50_ = 249.0 µM) on NCI-N87 gastric cells and as D3R (102.5 µM) > CA (207.7 µM) > EC (270.2 µM) on Caco-2 cells, respectively. In general, the lower IC_50_ values observed on NCI-N87 cells suggest that these polyphenols have stronger antiproliferative effects towards the gastric cancer-cell model used, when compared with the intestinal cancer model used (Caco-2).

### 2.2. Antiproliferative Activity of Binary Combinations of D3R, EC, and CA

The three compounds studied, D3R, EC, and CA, individually had a dose-dependent growth suppression in proliferating NCI-N87 and Caco-2 cells, therefore, all possible binary combinations of the three compounds were assessed, namely D3R+CA, D3R+EC, and CA+EC. All combinations acted in a dose-dependent manner on both cancer cells, decreasing cell viability (%) with an increase in the dose (Figure 1b). The polyphenols combination as well as the doses strongly affected the proliferation of NCI-N87 cells. For example, D3R+EC and D3R+CA appear as less effective than CA+EC to inhibit cell proliferation at lower doses, whereas at higher doses the opposite is observed (Figure 1b). At the intestinal level, the three combinations similarly influenced the viability of the Caco-2 cell model.

By comparing the dose-response graphs obtained from isolated compounds and binary mixtures, it is possible to infer the effect of compound combinations, notwithstanding the fact that the Chou-Talalay method provides algorithms for automated quantitative computer simulation for synergism and/or antagonism at any effect and dose level, as well as its graphical representation. Figure 2 and Figure 3 display the effect-oriented graphs (CI-plots) and dose-oriented graphs (isobolograms), respectively, giving the information about polyphenols interactions (synergism, additive, or antagonistic) towards NCI-N87 and Caco-2 proliferation, respectively. The interactions differed concerning the exposed concentrations: D3R+EC was the only combination with the same behaviour on both cells having an antagonistic effect at the low doses (CI > 1) and a synergetic effect (CI < 1) at the highest ones (Figure 2 and Figure 3). The other two combinations (D3R+CA and CA+EC) were synergistic at low levels and additive/antagonistic at the highest ones on gastric cells (particularly evident for the EC+CA combination), exhibiting opposite behaviour on Caco-2 cells. Thus, CA+EC was the combination with the strongest antiproliferative effect on NCI-N87 cells expressing strong synergism at low doses (until IC_25_), but having nearly additive and antagonistic effects at the highest inhibition concentration (>IC_80_) (Table 2 and Figure 2). In the case of Caco-2 cells, the aforementioned combination had the opposite behaviour (Table 2 and Figure 3).

Synergistic combinations may enable dose reduction, which can be advantageous. The DRI values provided using the chosen method are a measure of how-many-fold the dose of each compound in a synergistic combination may be reduced at a given effect level compared with the doses of each compound alone [15]. In the above referred synergistic interactions, combining polyphenols required a 1.7 to 15.3 times fewer amount of compounds to achieve a certain cell-viability level when compared with their individual exposure (Table 2). These values show the strength of the combinations, for instance, the IC_25_ and IC_50_ values of CA+EC combination on NCI-N87 cells were achieved using approximately 12–15 times and six times fewer amounts than in the individual exposure, respectively. On the other hand, on antagonistic interactions, namely D3R+CA and D3R+EC on Caco-2 cells at IC_25_ values, higher (31.5 µM) or similar (24.9 µM) amounts of D3R individually exposed (25.9 µM) were needed. Additionally, higher contents of EC (74.7 µM) were also required to inhibit 25% of Caco-2 cells along with D3R. This shows that the type of polyphenols and their doses in mixtures strongly influence their anticancer properties in gastric and intestinal cells.

## 3. Discussion

This study used human-derived gastric and colon cancer cells (NCI-N87 and Caco-2) as cellular models to understand the antiproliferative effect of three distinct chemical groups of polyphenols (anthocyanins—D3R, phenolic acids—CA, and flavanols—EC), together as binary combinations. The anthocyanin D3R was the compound with the highest antiproliferative effect on both cell lines exhibiting the lowest IC_50_ values (24.9 µM and 105.6 µM for gastric and intestinal cells, respectively) (Table 1). In this sense, blackcurrant berries (*Ribes nigrum* L.) present promising preventive effects against gastric and colon cancer cells proliferation because D3R is the most representative anthocyanin in those berries containing on average 250 mg of polyphenol compounds/100 g of fresh fruit [10]. Flavanols are also acknowledged for their anticancer properties as denoted for EC which exhibited the second lowest IC50 against cell proliferation [22].

To the best of our knowledge, no other studies were performed using the same cell lines and the Chou-Talatay theorem to allow comparing dose-response shapes for these compounds isolated or in combination. The increase in data of the interactions occurring amongst these compounds, targeting different physiological processes, provides an important tool for a better understanding of how human diets containing complex mixtures may exert chemopreventive or other beneficial effects [12]. Although complex, this knowledge of the phytochemicals’ interaction together with the comprehensive bioactive compound-food composition tables motivate the development of tools for dietary intervention targeting specific health effects.

Most of the studies usually expose polyphenol-rich extracts to cells rather than pure substances to evaluate their antiproliferative/anticancer effect [10,11,23], containing a complex mixture of compounds, making it difficult to identify synergism/antagonisms. Therefore, the present study used pure substances to understand the interactions between the selected polyphenols and the truly combined health effects. Synergetic effects against colon cancer have been reported by combining pure substances, such as resveratrol and curcumin [24], epigallocatechin-gallate (ECG) and EC [25], or combining curcumin along with 5-fluorouracil, as chemopreventive strategies [26].

The present study shows the interactions (synergisms/antagonisms) between D3R, CA, and EC to give knowledge of the best combinations to improve their bioactivity. Herein, synergetic, additive, and antagonistic effects were observed on all binary combinations on both cancer cells, being determined by the inhibition concentration and polyphenol dose tested (Table 2, Figure 2 and Figure 3). These results confirm that gastric and intestinal epithelium present variable susceptibility to polyphenols and their doses, which is in accordance with the findings of other authors [27,28].

In the case of NCI-N87 cells, lower contents of CA, D3R, and EC were required (from a 2.4 to 15.3-times-less amount) in D3R+CA and CA+EC combinations to inhibit proliferation until IC_75_ (Figure 2 and Table 2), compared with the amount needed on individual exposure. Interestingly, the combinations including CA were the ones with the strongest synergisms, which is relevant since this polyphenol had the lowest individual antiproliferative effect on gastric cells, which stresses the need to study the health effects of polyphenols in the light of its combinatory effects. Moreover, it is important to underline that the same mixture may be synergic at low doses but additive or antagonistic at the highest ones, which happened when combining D3R+CA on NCI-N87 cells (Figure 2). In this specific case, to synergistically inhibit 50% of the NCI-N87 cells proliferation, the optimal dose combinations would be 10.3 + 103.5 µM (D3R+CA) (Table 2). On the other hand, the D3R+EC combination had the opposite behaviour, shifting from slightly antagonistic at low doses to slightly synergistic at the highest ones.

Concerning Caco-2 cells, synergisms were only observed at a higher inhibition concentration (>IC_75_), requiring from 1.7 to 6.7-times-less the amount of compounds for each respective combination, except CA+EC, that was synergistic (CI = 0.66) at IC_50_ values (Table 2). This means that these specific polyphenols combinations are less active at low doses against colon cancer, being synergistic only when doses reach 94.4 + 188.9 µM (D3R+CA), 76.8 + 230.5 µM (D3R+EC), and 63.8 + 95.7 µM (CA+EC) (Table 2).

Phan, Paterson, Bucknall, and Arcot [15] recently reported several antagonistic interactions between anthocyanins (e.g., dephinidin-3-glucoside and pelargonidin-3-glucoside), anthocyanins and flavanols (e.g., delphiniding-3-glucoside and quercetin/querceting-3-glucoside), and flavonols and phenolic acids (e.g., catechin and ellagic acid). The loss of antioxidant response of the mixtures was justified by the formation of intermolecular hydrogen bonds between two different compounds reducing the availability of the active hydroxyl group for radical scavenging activities [15,18]. The high chemical interactivity of anthocyanins, due to their specific pyrylium nucleus (C-ring), can also justify the antagonistic effects observed [29]. Besides their weak acid properties, hard and soft electrophiles, and nucleophiles, the planar structures and extended electron delocalization over the three rings make the anthocyanins prone to develop π-stacking interactions, namely with other polyphenols [30,31]. The π-stacking interactions can be increased due to the acylation of anthocyanins on their glycosyl moieties via hydroxycinnamic acid residues [31]. D3R belongs to the class of anthocyanidin-3-o-glycosides, thus, the occurrence of chemical interactivity between D3R and CA (a hydroxycinnamic acid) and EC (a flavonoid) can contribute to the antagonism observed on Caco-2 cells for the combinations D3R+CA and D3R+EC at IC_25_. Interestingly, the same was not observed on NCI-N87 cells, which can be justified because the Chou-Talalay theorem is undertaken at equipotent contributions (i.e., (IC_50_)_1_/(IC_50_)_2_) with D3R presenting a higher antiproliferative activity in gastric cells; consequently the ratios of the compounds in the mixture are very different (with a smaller contribuition of D3R mass), thus the chemical interactivity was less noted (Table 1 and Table 2).

In conclusion, the interactions between D3R, CA, and EC herein described are biologically relevant, since their content and co-occurrence ratios in foods/supplements may influence the prevention and treatment of diseases, as demonstrated in vitro for the antiproliferative effect. Moreover, this study shows that the same combination causes different interactions/effects depending on the dose and cells; for example, synergistic effects were observed on the gastric cancer model at a maximum reduction dose of 15.3 times for CA in the CA/EC combination when polyphenols were at low doses, while on colon cancer cells only higher doses had synergistic effects reaching a maximum dose reduction effect of 6.4 times for EC in the same combination. It is thus of critical importance to accelerate the comprehensive discovery of the most abundant food bioactive-molecule interactions, in diverse physiological processes, and their compilation in organized databases. This information, together with phytochemical food contents, which are being gathered in very complete databases (e.g., https://foodb.ca and http://phenol-explorer.eu, accessed on 21 January 2023), play a relevant role in the determination of smart combinations of foods/extracts adjusted to potentiate a certain health effect.

## 4. Materials and Methods

### 4.1. Reagents and Materials

Delphinidin-3-*O*-rutinoside standard (D3R, ≥95% purity) was purchased from Extrasynthese (Genay, France). Chlorogenic acid (CA, >98% purity), epicatechin (EC, >90% purity), dimethyl sulfoxide (DMSO), 0.4% trypan blue stain solution, and MTT (3-(4,5-dimethylthiazol-2-yl)-2,5-diphenyltetrazolium bromide) were purchased from Sigma-Aldrich (St Louis, MO, USA). Fetal bovine serum (FBS), Roswell Park Memorial Institute medium (RPMI-1640), High glucose Dubelcco’s Modified Eagles Medium (DMEM), minimum essential medium non-essential amino acids (MEM NEAA) 100×, GlutaMAX^TM^ 100×, penicillin/streptomycin 100× solution (10.000 Units mL^−1^/10.000 µg mL^−1^), and 0.25% Trypsin-EDTA solution were purchased from Gibco/Life technology corporation (Paisley, UK).

### 4.2. Preparation of Standards Solutions

Stock solutions of D3R (20 mM) and CA (100 mM) were prepared in deionized water containing 10% DMSO, while EC (100 mM) was dissolved in methanol and stored at −80 °C until the preparation of the subsequent solutions. Working solutions of each standard were prepared in complete cell-culture medium (CM) before each experiment, ranging from 1.6 to 200 µM for D3R, 25 to 600 µM for EC, and 50 to 1000 µM for CA.

### 4.3. Cell Culture

Human-derived gastric cancer cell line (NCI-N87) was purchased from the American Type Culture Collection (ATCC) trough LGC standards (Barcelona, Spain), while the human epithelial colorectal adenocarcinoma cells (Caco-2 cells) were provided by the “Molecular Physical-Chemistry” group of the University of Coimbra, Portugal. NCI-N87 cells were grown in 75-cm^2^ culture flasks in CM constituted using RPMI-1640 with 10% heat-inactivated and 1% penicillin/streptomycin and were incubated at 37 °C with 5% CO_2_. In the case of Caco-2 cells, these were grown in 75-cm^2^ culture flasks in CM constituted using DMEM with 10% heat-inactivated FBS, 1% non-essential amino acids (NEAA, 1% glutaMAX, and 1% penicillin/streptomycin and incubated at 37 °C with 5% CO_2_. When the cells were approximately 80% confluent, they were trypsinized and seeded in 96-well plates (TPP, Trasadingen, Switzerland) to perform the antiproliferative assays.

### 4.4. Isolated and Combined Effects

The evaluation of the isolated antiproliferative activity of selected phenolic compounds on both cell lines (NCI-N87 and Caco-2) was performed by exposing the cells to at least 6 different concentrations, using different concentration ranges concerning their different antiproliferative effects towards cells. D3R concentrations ranged from 6.3 to 200 µM (NCI-N87 cells) and 1.6 to 200 µM (Caco-2 cells); EC concentrations from 25 to 400 µM (NCI-N87 cells) and 25 to 600 µM (Caco-2 cells); and finally, CA concentrations ranged from 50 to 1000 µM (NCI-N87 cells) and 100 to 1000 µM (Caco-2 cells) to obtain a good fitting of dose-response curves. The concentration which inhibits half of the cells proliferation (IC_50_) of each isolated compound was calculated from the respective dose-response curve for both cell lines. The cytotoxicity of combined compounds was studied in binary combination, D3R+CA, D3R+EC, and CA+EC, by exposing the cells to 5 dilutions of the mixtures following constant ratio combination design developed by the Chou-Talalay method. This method was carried out at an equipotency ratio (i.e., (IC_50_)_1_/(IC_50_)_2_ ratio) so that the contributions of each compound effect to the combination would be equal [19].

### 4.5. Cell Viability Assay

The isolated and combined effects of CA, EC, and D3R were evaluated with MTT assay. Proliferating NCI-N87 (2500 cells/ well) and Caco-2 cells (1250 cells/ well) were seeded in 96-well plates at 37 °C, 5% CO_2_ for 24 h, allowing the cells to adhere on the surface of the wells. After that, cells were exposed for 72 h to D3R, CA, and EC firstly alone, and after in combination. Wells treated with CM alone and with maximum methanol (1%) or DMSO content (0.1%) on CM solution were used as controls to ensure the viability of cells during the experiments.

### 4.6. Data Analyses

All data are presented as mean ± standard deviation in cell viability graphs from at least three independent experiments (quadruplicate). GraphPad Prism software was used to build the dose-effect curves of each standard (Graphpad Software version 5.00 for windows, San Diego, CA, USA). The software Compusyn version 1.0 was used to calculate the IC_50_ value of isolated compounds as well as the combination index (ComboSyn Inc., Paramus, NJ, USA). This software uses the median-effect equation of the mass-action law, which considers the shape of the dose-response curve, either when m = 1 hyperbolic, m > 1 sigmoidal, m < 1 flat sigmoidal [19,32]:(1)$${\log\;(f}_{a}{/f}_{u})=mlogD\;-{\;mlogD}_{m},$$
where *D* is the dose, *D_m_* is the IC_50_, *f_a_* is the fraction affected by the dose *D*, *m* is the coefficient of the sigmoidicity of the dose effect, and *f_u_* = 1 − *f_a_*.

The combination index (CI) theorem of Chou and Talatay [32] developed for quantification of synergism, additive effect, or antagonism for two drugs was used to determine the effect of the binary combinations of the selected phenolic compounds on cells:(2)$$CI=\frac{{\left(D\right)}_{1}}{{\left(D_{m}\right)}_{1}}+\frac{{\left(D\right)}_{2}}{{{(D}_{m})}_{2}},$$
where (*D_m_*)_1_ and (*D_m_*)_2_ are the doses of individual compounds that correspond to the IC_50_. (*D*)_1_ and (*D*)_2_ are the doses of the two phenolic compounds that combined inhibited half of the cells growth (IC_50_). When *CI* is around 1 (0.9 < *CI* < 1.1), the combination of phenolic compounds has additive effect on cells. If *CI* < 0.9, the combination has synergistic effect on cells and if *CI* > 1.1, it possesses antagonistic effect. *CI* values were expressed as CI-Plot and isobologram graphs. This software also calculates dose reduction indices (*DRI*) for synergistic combinations. *DRI* values can be obtained from the reciprocal of *CI* equation:(3)$${\left(DRI\right)}_{1}=\frac{{\left(D_{x}\right)}_{1}}{{\left(D\right)}_{1}}\;\mathrm{and}{\;(DRI)}_{2}=\frac{{\left(D_{x}\right)}_{2}}{{\left(D\right)}_{2}}$$
where (*D_x_*)_1_ and (*D_x_*)_2_ are the doses that the standards alone inhibited x% of cells, and (*D*)_1_ and (*D*)_2_ are the values that standards combined inhibited x% of cells and show a dose reduction rate of each compound in the combination that achieves the same inhibition compared with the doses of each compound alone.

## Figures and Tables

**Figure 1 molecules-28-01286-f001:**
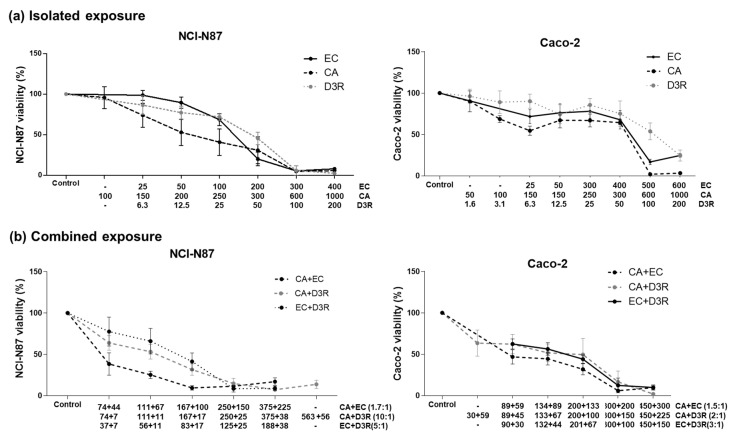
Viability (%) of proliferating NCI-N87 and Caco-2 cells exposed to dilutions of EC, CA, and D3R (µM) for 72 h, (**a**) isolated and (**b**) in binary combination (CA+EC, CA+D3R, and EC+D3R). Data are presented as mean ± SD of at least two independent experiments.

**Figure 2 molecules-28-01286-f002:**
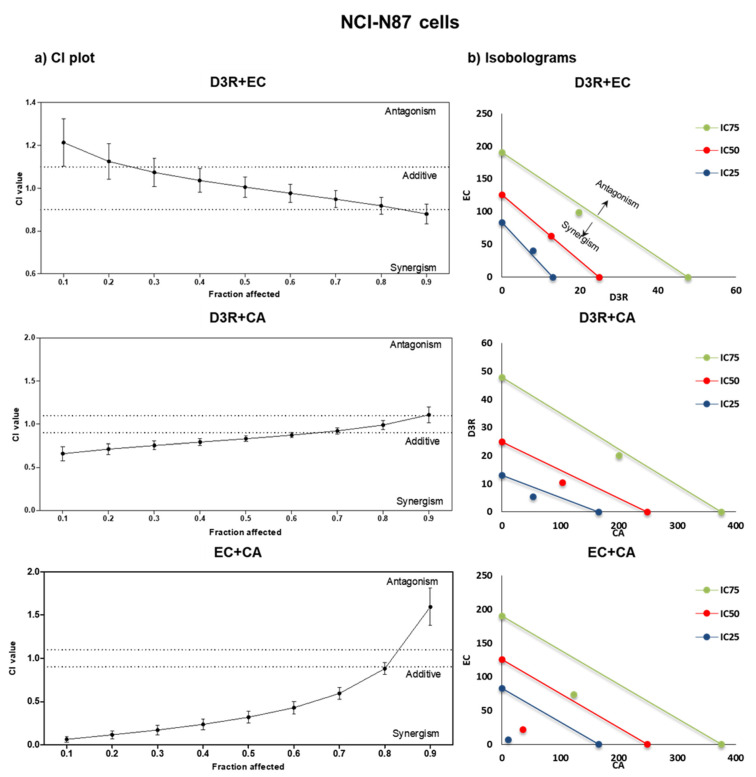
CI-plots (**a**) and isobolograms (**b**) of D3R, CA, and EC binary combinations: EC+D3R, CA+D3R, and CA+EC after exposure on NCI-N87 cells. The CI values were obtained from three independent experiments (quadruplicates); the vertical bars on CI-plots indicate 95% confidence intervals for CI values based on sequential deletion analysis [19].

**Figure 3 molecules-28-01286-f003:**
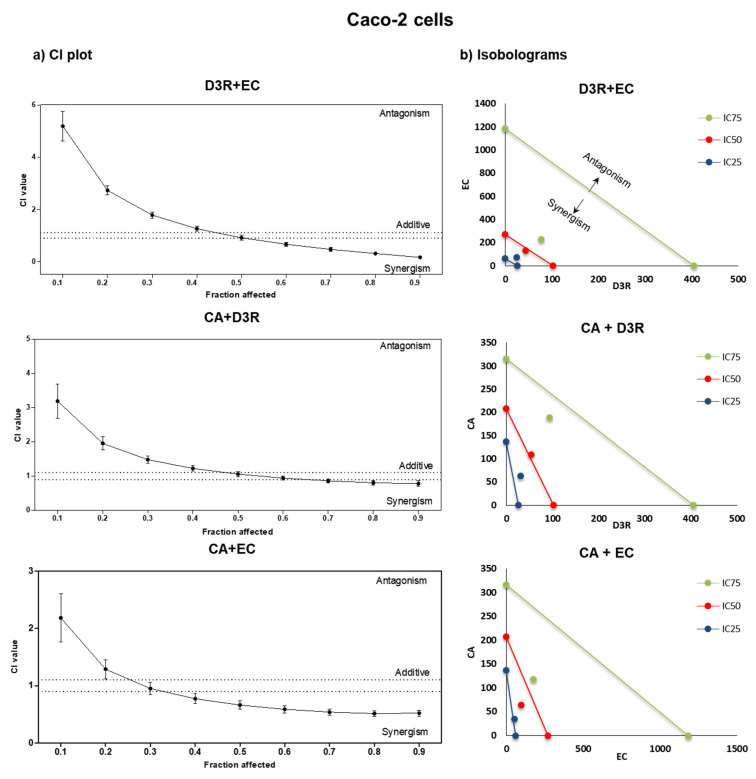
CI-plots (**a**) and isobolograms (**b**) of D3R, CA, and EC binary combinations: D3R+EC, D3R+CA, and CA+EC after exposure on Caco-2 cells. The CI values were obtained from three independent experiments (quadruplicates); the vertical bars on CI-plots indicate 95% confidence intervals for CI values based on sequential deletion analysis [19].

**Table 1 molecules-28-01286-t001:** Dose-effect parameters for individual exposure of polyphenols on NCI-N87 and Caco-2 cells.

(a) NCI N87 Cells					
	r	m	IC25	IC50 (D_m_)	IC75
D3R	0.96	1.69	13.0	24.9	47.8
CA	0.93	2.67	165.1	249.0	375.4
EC	0.98	2.65	83.3	126.1	190.8
(b) Caco-2 cells					
	r	m	IC25	IC50 (D_m_)	IC75
D3R	0.86	0.79	25.9	102.5	405.3
CA	0.87	2.61	136.4	207.7	316.3
EC	0.79	0.74	61.6	270.2	1184

r values represent the goodness of the fit curve, m signifies the shape of the dose-effect curve (m = 1, hyperbolic, m > 1 sigmoidal, m < 1 flat sigmoidal) and Dm represents median-effect dose (in this case the IC50 values). All inhibition concentrations (ICx) are expressed as µM. D3R—Delphinidin-3-rutinoside, EC—epicatechin, CA—chlorogenic acid.

**Table 2 molecules-28-01286-t002:** Dose-effect parameters (r, m, D_m_), combination index (CI), and dose reduction index (DRI) values for combinations of polyphenols towards NCI-N87 and Caco-2 cells proliferation. Dose-effect parameters for individual exposure of polyphenols on NCI-N87 and Caco-2 cells.

NCI-N87 Cells												
	Combination Ratio	r	m	IC_25_	CI	DRI	IC_50_	CI	DRI	IC_75_	CI	DRI
D3R	10:1	0.93	1.66	5.3	0.73	3.1	10.3	0.83	2.4	20.0	0.93	
CA				53.5		2.4	103.5		2.4	200.0		
D3R	5:1	0.93	2.43	8.0	1.09		12.6	1.0		19.8	0.95	
EC				40.1			63.1			99.1		
CA	1.7:1	0.71	0.9	10.8	0.14	15.3	36.4	0.32	6.8	122.9	0.71	3.1
EC				6.48		12.9	21.9		5.8	73.7		2.6
**Caco-2 cells**												
	Combination ratio	r	m	IC_25_	CI	DRI	IC_50_	CI	DRI	IC_75_	CI	DRI
D3R	2:1	0.84	2.0	31.5	1.68		54.6	1.05		94.4	0.83	1.7
CA				63.1			109.1			188.9		4.3
D3R	3:1	0.93	1.95	24.9	2.17		43.7	0.91		76.8	0.38	5.1
EC				74.7			131.2			230.5		5.3
CA	1.5:1	0.87	1.78	34.4	1.09		63.8	0.66	3.3	118.5	0.52	2.7
EC				51.6			95.7		2.8	177.7		6.7

Description of synergism/antagonism based on CI values: strong synergism (0.1–0.3); synergism (0.3–0.7); moderate synergism (0.7–0.85); slight synergism (0.85–0.9); nearly additive (0.9–1.0); slight antagonism (1.1–1.2); moderate antagonism (1.2–1.45); antagonism (1.45–3.3); and strong antagonism (3.3–10) [19]. r values represent the goodness of the fit curve; m signifies the shape of the dose-effect curve (m = 1, hyperbolic, m > 1 sigmoidal, m < 1 flat sigmoidal). All inhibition concentrations (ICx) are expressed as µM.

## Data Availability

Not applicable.

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
