# Peer review of "Delphinidin-3-rutinoside from Blackcurrant Berries (Ribes nigrum): In Vitro Antiproliferative Activity and Interactions with Other Phenolic Compounds"

_molecules, 2023, doi:10.3390/molecules28031286_

Round 1

Reviewer 1 Report (Previous Reviewer 2)

The manuscript has been revised fully following my comments. I recommend its acceptance for publication.

Author Response

Authors thank the reviewer recommendation to publish the paper.

Reviewer 2 Report (New Reviewer)

- The authors should check the final version of the manuscript before any submission;

- The authors could try tertiary combinations to get more information about the synergistic, additive or antagonistic effect of D3R with other tested phenolic compounds.

- Why the authors did choose 72h as time of incubation? They could try different times of exposure: 24h, 48h,...to get more information.

-Update the references about works done on blackcurrant berries;

- It would be interesting as perspectives to study the effect of delphinidin-3-rutinoside freshly extracted from blackcurrant berries to confirm those results and also make recommendations or suggestions about the amount of the fruit that could be included in the diet to reduce the risk of gastric and intestinal cancers;

Author Response

- The authors should check the final version of the manuscript before any submission.

The final version of the manuscript was checked and confirmed.

- The authors could try tertiary combinations to get more information about the synergistic, additive or antagonistic effect of D3R with other tested phenolic compounds.

Authors understand the reviewer point of view, however, the combination of three drugs is not an advantage, mainly due to two reasons:

  • In the Chou-Talalay method, three drugs are considered as if they were two drugs (https://doi.org/10.1124/pr.58.3.10). In this experimental design, we have the dose-effect curves for D3R, CA, and EC alone, for two-drug combinations D3R+CA, D3R+EC, CA+EC. The two-drug combination dissect the three-drug combinations and provide more insight into the interactions of their components which was accomplished in the present work. The evaluation of the three drug combinations does not need the previous 2 drugs combinations assessment, however, as said, the two-drug combination provide a more comprehensive evaluation of the compounds interactions;
  • To avoid experimental variability due to variables in assay conditions, it is recommended that dose-effect curves for each single drug and its combinations be carried out at the same time, as stated in the method recommendations (https://doi.org/10.1124/pr.58.3.10). Due to the number of cell lines (2 ) and the high range of concentrations used (6 points on average) the practical execution of all the assays in the same time-frame is not possible, reason why binary combinations can provide us more information than tertiary ones.

- Why the authors did choose 72h as time of incubation? They could try different times of exposure: 24h, 48h,...to get more information.

The exposure time of 72h was selected as an approximation to the exposure obtained from dietary sources, i.e. low doses and multiple exposition. To make the same amount of assays and obtain the amount of information for two cell types, the range of concentration and replicates used, is an amount of work that, in our opinion, largely surpasses the usual amount of work required for a scientific paper of this type.

-Update the references about works done on blackcurrant berries;

Reference list was updated, the following references were inserted:

  1. Liu, B.; Li, Z. Black Currant (Ribes nigrum L.) Extract Induces Apoptosis of MKN-45 and TE-1 Cells Through MAPK- and PI3K/Akt-Mediated Mitochondrial Pathways. Med. Food 2016, 19, 365-373, doi:10.1089/jmf.2015.3521.
  2. Koss-Mikołajczyk, I.; Kusznierewicz, B.; Bartoszek, A. The Relationship between Phytochemical Composition and Biological Activities of Differently Pigmented Varieties of Berry Fruits; Comparison between Embedded in Food Matrix and Isolated Anthocyanins. Foods 2019, 8, doi:10.3390/foods8120646.
  3. Phan, M.A.T.; Paterson, J.; Bucknall, M.; Arcot, J. Interactions between phytochemicals from fruits and vegetables: Effects on bioactivities and bioavailability. Rev. Food Sci. Nutr. 2018, 58, 1310-1329, doi:https://doi.org/10.1080/10408398.2016.1254595.
  4. Borowiec, K.; Stachniuk, A.; Szwajgier, D.; Trzpil, A. Polyphenols composition and the biological effects of six selected small dark fruits. Food Chem. 2022, 391, 133281, doi:https://doi.org/10.1016/j.foodchem.2022.133281.

- It would be interesting as perspectives to study the effect of delphinidin-3-rutinoside freshly extracted from blackcurrant berries to confirm those results and also make recommendations or suggestions about the amount of the fruit that could be included in the diet to reduce the risk of gastric and intestinal cancers.

The reviewer suggestion is interesting however the delphinidin-3-rutinoside used is a pure standard obtained from blackcurrant berries acquired at the company Extrasynthese (https://www.extrasynthese.com/delphinidin/270-delphinidin-3-o-rutinoside-chloride.html) thus is equivalent to extract the compounds from fresh fruits.

Concerning the amount of fruit that could be included in the diet to reduce the risk of gastric and intestinal cancer we do not have enough data at the present to make such a suggestion. For instance, the compound transformation during gastrointestinal digestion is not well known. Moreover, the content of delphinidin-3-rutinoside is variable between blackcurrants (production location, variety, etc) which difficult recommendations for its consumption. In addition, as stated in lines 64-67, the doses at which these compounds are present in diet, as well as the presence of other polyphenols, may also determine the interaction (synergism/antagonism). Notwithstanding, our data is relevant to properly design specific and innovative functional/bioactive foods since is a pioneer work on interactions of delphinidin-3-rutinoside.

Reviewer 3 Report (New Reviewer)

The manuscript presented a very promising investigation, well presented, but with few recent publication on the topic.

Minor correction:

-Line 171: position of Figure 2 legend.

-Please improve Table 2.

-Please upgrade reference list

Author Response

The manuscript presented a very promising investigation, well presented, but with few recent publications on the topic.

We thank the reviewer nice words. Concerning the reference to a few recent publications on the topic of interactions of bioactive compounds (in this case anthocyanins, flavanols, and hydroxycinnamic acids) there are no bibliographic records other than those indicated in the manuscript.

Minor correction:

-Line 171: position of Figure 2 legend.

The position of Figure 2 Legend was corrected.

-Please improve Table 2.

Table 2 position was corrected and improved.

-Please upgrade reference list

Reference list was updated, the following references were inserted:

  1. Liu, B.; Li, Z. Black Currant (Ribes nigrum L.) Extract Induces Apoptosis of MKN-45 and TE-1 Cells Through MAPK- and PI3K/Akt-Mediated Mitochondrial Pathways. Med. Food 2016, 19, 365-373, doi:10.1089/jmf.2015.3521.
  2. Koss-Mikołajczyk, I.; Kusznierewicz, B.; Bartoszek, A. The Relationship between Phytochemical Composition and Biological Activities of Differently Pigmented Varieties of Berry Fruits; Comparison between Embedded in Food Matrix and Isolated Anthocyanins. Foods 2019, 8, doi:10.3390/foods8120646.
  3. Phan, M.A.T.; Paterson, J.; Bucknall, M.; Arcot, J. Interactions between phytochemicals from fruits and vegetables: Effects on bioactivities and bioavailability. Rev. Food Sci. Nutr. 2018, 58, 1310-1329, doi:https://doi.org/10.1080/10408398.2016.1254595.
  4. Borowiec, K.; Stachniuk, A.; Szwajgier, D.; Trzpil, A. Polyphenols composition and the biological effects of six selected small dark fruits. Food Chem. 2022, 391, 133281, doi:https://doi.org/10.1016/j.foodchem.2022.133281.

This manuscript is a resubmission of an earlier submission. The following is a list of the peer review reports and author responses from that submission.

Round 1

Reviewer 1 Report

Compliments  to the authors on a very important work. The study is valuable by highlighting the individual effect and the relationships established between the three polyphenols and the doses in which they are effective against the two cancer models.

The only suggestion would be to revise the order in which two polyphenols from the D3R combinations are mentioned both in the legend and next to the oX axis, in figure 1. In any case, this aspect does not hinder the understanding of the study.

Author Response

Comment: Q1 - Compliments to the authors on a very important work. The study is valuable by highlighting the individual effect and the relationships established between the three polyphenols and the doses in which they are effective against the two cancer models.

Answer: Authors acknowledge reviewer compliments and comments as we share the same opinion on the importance of the submitted work as a more comprehensive approach to bioactivity studies.

Comment: The only suggestion would be to revise the order in which two polyphenols from the D3R combinations are mentioned both in the legend and next to the oX axis, in figure 1. In any case, this aspect does not hinder the understanding of the study.

Answer: Figure 1 was modified accordingly to reviewer comments.

Reviewer 2 Report

The manuscript entitled 'Delphinidin-3-rutinoside from blackcurrant berries (Ribes nigrum): in vitro antiproliferative activity and interactions with other phenolic compounds' is very interesting and has good scope to help the scientific community in understanding the in vitro modles of testing polyphenols as nutraceuticals. I have appended the reviewer PDF report. 

Reviewer 3 Report

Miladinovic et al. have studied the anti-proliferative effect of phenolic compounds, phenolic compounds, such as, delphinidin-3-rutinoside (D3R) and chloro- 15 genic acid (CA) and epicatechin (EC). These compounds are purchased from commercial vendors and are known to be found in blackcurrant berries (Ribes nigrum). Two human cell lines NCI-N87 (gastric) and Caco-2 (intestinal) cells are used to analyze the antiproliferative effect of these compounds individually and in binary combination. The synergism and antagonism of these compounds is studied. Manuscript provides the following conclusion. 1. D3R showed the strongest antiproliferative activity among all three molecules tested. 2. The combinations, D3R+CA and CA+EC show synergic effect at lower dose against NCI-N87 proliferation, and 3. Antagonist effect at lower dose and synergic effect at higher dose against Caco-2 proliferation. Overall, the work presents quite a narrow experimental design and results that these compounds D3R, CA and EC have antiproliferative effects. Due to the lack of completeness, the manuscript is not recommended for publication in its present form. My specific comments are as follows:

1.       The experimental design used in this work is very limited. Results of which only show the effect of selected compounds on proliferation of selected cell lines. It lacks, first of all, control that is the effect of this compound on non-cancerous cell lines; secondly, the chemical interaction of this compound with each other; third, other (than MTT assay) evidences showing anti-proliferative effects. It is difficult to conclude the synergism and antagonism of used compounds without knowing their chemical interactivity with each other.

2.       Several questions are left unanswered in the paper.

·        What is the possible mode of action behind the antiproliferative effect D3R, CA and EC.

·        Are these compounds potential candidates for anti-cancer drugs?

·        How are they more/less effective compared to other known compounds of similar chemical nature?

·        Why do they have different activities (synergism/antagonism) at different doses?    

3.       The title of the manuscript highlights “blackcurrant berries (Ribes nigrum)”. Are those compounds (D3R, CA and EC) directly extracted from blackcurrant berries OR chemically synthesized? This should be clarified. If they are chemically synthesized, the use of the word “blackcurrant berries (Ribes nigrum)” in the title brings skepticism.  

Author Response

Comment: Overall, the work presents quite a narrow experimental design and results that these compounds D3R, CA and EC have antiproliferative effects. Due to the lack of completeness, the manuscript is not recommended for publication in its present form. My specific comments are as follows:

Q1:  The experimental design used in this work is very limited. Results of which only show the effect of selected compounds on proliferation of selected cell lines. It lacks, first of all, control that is the effect of this compound on non-cancerous cell lines; secondly, the chemical interaction of this compound with each other; third, other (than MTT assay) evidences showing anti-proliferative effects. It is difficult to conclude the synergism and antagonism of used compounds without knowing their chemical interactivity with each other.

Answer - Authors acknowledge reviewer comments, to answer to the raised questions is as follows: The chemical interactions of the compounds with each other were tested following the Chou-Talalay method evaluating their antiproliferative activity (MTT to measure cell viability) as an end-point. The method to evaluate compounds combination is based on the median-effect equation, derived from the mass-action law principle. This general equation encompasses the Michaelis-Menten, Hill, Henderson-Hasselbalch, and Scatchard equations thus its application is highly broad and there is no need to know previously the mechanisms of action (https://doi.org/10.1158/0008-5472.can-09-1947).

We agree with the reviewer concerning the usefulness of testing the synergisms in non-cancerous cell lines however, we consider that approach as a second step of the in vitro analysis, since it was not applicable in a single work due to the huge amount of work needed.

The use of another assay to test the antiproliferative activity could be considered though, the MTT test, due to its simplicity and high-throughput, was selected since a very high amount of assays is needed to determine, in vitro, firstly the individual dose-response effects and then all the combinations of compounds. Another reason to use one single end-point, the MTT assay, in this case, is concerned with the fact that the Chou-Talalay method does not allow the combinations of results from different end-points. In sum, the use of another end-point to measure the antiproliferative effect would be impracticable with the followed methodology.

Q2.2 Are these compounds potential candidates for anti-cancer drugs?

The compounds were evaluated concerning its effects on gastric and intestinal cancer cells proliferation envisaging its putative use in food supplements and smart food synergistic combinations of the food products containing them. The in vitro evaluation if the first and necessary step in the identification of the synergistic combinations. The information provided by the present study can be used in further in vivo studies, e.g. animal studies, human dietary interventions, etc. Moreover, this information, together with phytochemicals food contents, which are being gathered in very complete databases (e.g. https://foodb.ca and http://phenol-explorer.eu), plays a relevant role in the determination of smart combinations of foods/extracts adjusted to potentiate a certain health effect.

Q2.3   How are they more/less effective compared to other known compounds of similar chemical nature?

The present work used two cell lines as models of gastrointestinal cancer, the gastric NCI-N87 and the intestinal Caco-2. For the NCI-N87 model, to the best of our knowledge, no antiproliferative effects of anthocyanins, phenolic acids and flavanols, as pure compounds, were tested or reported, thus comparison with other compounds of similar nature is not feasible. For the Caco-2 model, a few studies report the antiproliferative activity of these compounds families however not in the same conditions (mainly determined by the time of exposure, 72 hours) reason why we did not include any reference for these results, as they are not directly compared. For instance, Martini and collaborators (2019) reported no effect of p-coumaric, o-coumaric, cinnamic, and 5-caffeoylquinic acids at 200 μM after treatment for 24 h on Caco-2 cells (https://pubs.acs.org/doi/10.1021/acs.jafc.9b00522). Concerning anthocyanins, a reduction of about 50% in cell proliferation was noticed for cyanidin-3-glucoside at a concentration of 0.2 mg/ml (410 μM) after 12 h exposition (https://doi.org/10.1039/C7RA06387C). Delgado and colleagues (2014) reported no inhibitory activity on cell growth after 48 exposition to 100 µM EC but found an IC50 value of 85.21 µM for quercetin (https://doi.org/10.1039/C3FO60441A). Moreover, from our bibliographic searches most of the reported antiproliferative effects were evaluated in extracts, not pure compounds.

Q2.4 Why do they have different activities (synergism/antagonism) at different doses?   

The possibility to assess the different interactions at different doses is the main advantage of the Chou-Talalay method and is frequently determined in synergisms/antagonisms assessment. The method helps answer the following relevant primary questions: (a) Are there any synergisms? (b) How much synergism? (c) Synergism at what dose levels? (d) Synergism at what effect levels. If multiple data points for constant-ratio combination are available, the entire spectrum of synergism or antagonism at all effect levels can be automatically simulated and is presented herein.

Q3:  The title of the manuscript highlights “blackcurrant berries (Ribes nigrum)”. Are those compounds (D3R, CA and EC) directly extracted from blackcurrant berries OR chemically synthesized? This should be clarified. If they are chemically synthesized, the use of the word “blackcurrant berries (Ribes nigrum)” in the title brings skepticism. 

The compounds were obtained from the company Extrasynthese that extracts and purifies compounds from plants. In the case of D3R it was extracted from the blackcurrant Ribes nigrum (https://www.extrasynthese.com/delphinidin/270-delphinidin-3-o-rutinoside-chloride.html).

Reviewer 4 Report

This study investigated the human-derived gastric and colon cancer cells (NCI-N87 and Caco-2) as cellular models to understand the antiproliferative effect of 3 distinct chemical groups of polyphenols (anthocyanins - D3R, phenolic acids - chlorogenic acid and flavanols - epicatechin), together as binary combinations. The authors conclude that the same combination originates different interactions depending on dose and cells such as synergism effects were observed on the gastric cancer model when polyphenols were at low doses, while on colon cancer cells only higher doses polyphenol ratios had synergistic effects. Also the Delphinidin-3-rutinoside was the compound with the highest antiproliferative effect on both cell lines exhibiting the lowest IC50 values.

No other studies were done using the human-derived gastric and colon cancer cells (NCI-N87 and Caco-2) as cellular models and Chou-Talatay theorem to understand the antiproliferative effect of 3 distinct chemical groups of polyphenols. The conducted research allows comparing dose-response shapes for these compounds isolated or in combination.

The methods are adequately described, the results are clearly presented. Additionally, the conclusions are supported by the results and the figures are of high quality. Also, the references are appropriate.

I recommend the article to be published in its current form.

Author Response

Authors acknowledge the reviewer opinion and recommendation to publish in its current form.

Round 2

Reviewer 3 Report

Hi,

The answers/revisions to my comments are not satisfactory as it doesn't provide any added information like, chemical interactivity of tested compounds and other methods to check anti-proliferative activities. I wouldn't recommend the manuscript for publication in its present form. 

Thank you!